# DEEP LATENT STATE SPACE MODELS FOR TIME-SERIES GENERATION

## ABSTRACT

Methods based on ordinary differential equations (ODEs) are widely used to build generative models of time-series. In addition to high computational overhead due to explicitly computing hidden states recurrence, existing ODE–based models fall short in learning sequence data with sharp transitions – common in many real-world systems – due to numerical challenges during optimization. In this work, we propose LS4, a generative model for sequences with latent variables evolving according to a state space ODE to increase modeling capacity. Inspired by recent deep state space models (S4), we achieve speedups by leveraging a convolutional representation of LS4 which bypasses the explicit evaluation of hidden states. We show that LS4 significantly outperforms previous continuous-time generative models in terms of marginal distribution, classification, and prediction scores on real-world datasets in the Monash Forecasting Repository, and is capable of modeling highly stochastic data with sharp temporal transitions. LS4 sets state–of–the–art for continuous–time latent generative models, with significant improvement of mean squared error and tighter variational lower bounds on irregularly–sampled datasets, while also being $\times 100$ faster than other baselines on long sequences.

## 1 INTRODUCTION

Time series are a ubiquitous data modality, and find extensive application in weather (Hersbach et al., 2020) engineering disciplines, biology (Peng et al., 1995), and finance (Poli et al., 2019). The main existing approaches for deep generative learning of temporal data can be broadly categorized into autoregressive (Oord et al., 2016), latent autoencoder models (Chen et al., 2018; Yildiz et al., 2019; Rubanova et al., 2019), normalizing flows (de Bézenac et al., 2020), generative adversarial (Yoon et al., 2019; Yu et al., 2022; Brooks et al., 2022), and diffusion (Rasul et al., 2021). Among these, continuous-time methods (often based on underlying ODEs) are the preferred approach for irregularly-sampled sequences because they can make predictions at arbitrary time steps and can handle sequences of varying lengths. Unfortunately, existing ODE–based methods (Rubanova et al., 2019; Yildiz et al., 2019) often fall short in learning models for real-world data (e.g., with stiff dynamics) due to their limited expressivity and numerical instabilities during backward gradient computation (Hochreiter, 1998; Niesen & Hall, 2004; Zhuang et al., 2020).

A natural way to increase the flexibility of ODE-based models is to increase the dimensionality of their (deterministic) hidden states. However, that leads to quadratic scaling in the hidden dimensionality due to the need of explicitly computing hidden states by unrolling the underlying recurrence over time, thus preventing scaling to long sequences. An alternative approach to increasing modeling capacity is to incorporate *stochastic* latent variables into the model, a highly successful strategy in generative modeling (Kingma & Welling, 2013; Chung et al., 2015; Song et al., 2020; Ho et al., 2020). However, this leads to computational costs, and existing models like latent neural ODE models (Rubanova et al., 2019) inject stochasticity only at the initial condition of the system. In contrast, we introduce LS4, a latent generative model where the sequence of latent variables is represented as the solution of linear state space equations (Chen, 1984). Unrolling the recurrence equation shows an autoregressive dependence in the sequence of latent variables, the joint of which is highly expressive in representing time series distributions. The high dimensional structure of the latent space, being equivalent to that of the input sequence, allows LS4 to learn expressive latent representations and fit the distribution of sequences produced by a *family* of dynamical systems, a common setting

resulting from non–stationarity. We further show how LS4 can learn the dynamics of *stiff* (Shampine & Thompson, 2007) dynamical systems where previous methods fail to do so. Inspired by recent works on deep state space models, or stacks of linear state spaces and non-linearities (Gu et al., 2020; 2021), we leverage a convolutional kernel representation to solve the recurrence, bypassing any explicit computations of hidden states via the recurrence equation, which ensures log–linear scaling in both the hidden state space dimensionality as well as sequence length.

We validate our method across a variety of time series datasets, benchmarking LS4 against an extensive set of baselines. We propose a set of 3 metrics to measure the quality of generated time series samples and show that LS4 performs significantly better than baselines on datasets with stiff transitions and obtains on average $30\%$ lower MSE scores and ELBO. On sequences with $\approx 20K$ lengths, our model trains $\times 100$ faster than the baseline methods.

## 2 RELATED WORK

Rapid progress on deep generative modeling of natural language and images has consolidated diffusion (Ho et al., 2020; Song et al., 2020; Song & Ermon, 2019; Sohl-Dickstein et al., 2015) and autoregressive techniques (Brown et al., 2020) as the state–of–the–art. Although various approaches have been proposed for generative modeling of time series and dynamical systems, consensus on the advantages and disadvantages of each method has yet to emerge.

**Deep generative modeling of sequences.** All the major paradigms for deep generative modeling have seen application to time series and sequences. Most relevant to our work are latent continuous–time autoencoder models proposed by Chen et al. (2018); Yildiz et al. (2019); Rubanova et al. (2019), where a neural differential equation encoder is used to parametrize as distribution of initial conditions for the decoder. Massaroli et al. (2021) proposes a variant that parallelizes computation in time by casting solving the ODE as a root finding problem. Beyond latent models, other continuous–time approaches are given in Kidger et al. (2020), which develops a GAN formulation using SDEs.

**State space models.** *State space models.* (SSMs) are at the foundation of dynamical system theory (Chen, 1984) and signal processing (Oppenheim, 1999), and have also been adapted to deep generative modeling. Chung et al. (2015); Bayer & Osendorfer (2014) propose VAE variants of discrete–time RNNs, generalized later by (Franceschi et al., 2020), among others. These models all unroll the recurrence and are thus challenging to scale to longer sequences.

Our work is inspired by recent advances in deep architectures built as stacks of linear SSMs, notably S4 (Gu et al., 2021). Similar to S4, our generative model leverages the convolutional representation of SSMs during training, thus bypassing the need to materialize the hidden state of each recurrence.

## 3 PRELIMINARIES

We briefly introduce relevant details of continuous-time SSMs and their different representations. Then we introduce preliminaries of generative models for sequences.

### 3.1 STATE SPACE MODELS (SSM)

A *single-input single-output (SISO)* linear state space model is defined by the following differential equation

$$
\begin{aligned}
\frac{d}{dt}\mathbf{h}_t &= \boldsymbol{A}\mathbf{h}_t + \boldsymbol{B}x_t \\
y_t &= \boldsymbol{C}\mathbf{h}_t + \boldsymbol{D}x_t
\end{aligned}
\tag{1}
$$

with scalar *input* $x_t \in \mathbb{R}$, *state* $\mathbf{h}_t \in \mathbb{R}^N$ and scalar *output* $y_t \in \mathbb{R}$. The system is fully characterized by the matrices $\boldsymbol{A} \in \mathbb{R}^{N \times N}, \boldsymbol{B} \in \mathbb{R}^{N \times 1}, \boldsymbol{C} \in \mathbb{R}^{1 \times N}, \boldsymbol{D} \in \mathbb{R}^{1 \times 1}$. Let $x, y \in \mathcal{C}([a, b], \mathbb{R})$ be absolutely continuous real signals on time interval $[a, b]$. Given an initial condition $\mathbf{h}_0 \in \mathbb{R}^N$ the SSM (1) realizes a mapping $x \mapsto y$.

SSMs are a common tool for processing continuous input signals. We consider *single input single output* (SISO) SSMs, noting that input sequences with more than a single channel can be processed

by applying multiple SISO SSMs in parallel, similarly to regular convolutional layers. We use such SSMs as building blocks to map each input dimension to each output dimension in our generative model.

**Discrete recurrent representation.** In practice, continuous input signals are often sampled at time interval $\Delta$ and the sampled sequence is represented by $x = (x_{t_0}, x_{t_1}, \ldots, x_{t_L})$ where $t_{k+1} = t_k + \Delta$. The discretized SSM follows the recurrence

$$
\begin{aligned}
\mathbf{h}_{t_{k+1}} &= \bar{\boldsymbol{A}} \mathbf{h}_{t_k} + \bar{\boldsymbol{B}} x_{t_k} \\
y_{t_k} &= \boldsymbol{C} \mathbf{h}_{t_k} + \boldsymbol{D} x_{t_k}
\end{aligned}
\tag{2}
$$

where $\bar{\boldsymbol{A}} = e^{\boldsymbol{A}\Delta}$, $\bar{\boldsymbol{B}} = \boldsymbol{A}^{-1}(e^{\boldsymbol{A}\Delta} - I)\boldsymbol{B}$ with the assumption that signals are constant during the sampling interval. Among many approaches to efficiently computing $e^{\boldsymbol{A}\Delta}$, Gu et al. (2021) use a bilinear transform to estimate $e^{\boldsymbol{A}\Delta} \approx (I - \frac{1}{2}\boldsymbol{A}\Delta)^{-1}(I + \frac{1}{2}\boldsymbol{A}\Delta)$.

This recurrence equation can be used to iteratively solve for the next hidden state $h_{t_{k+1}}$, allowing the states to be calculated like an RNN or a Neural ODE (Chen et al., 2018; Massaroli et al., 2020).

**Convolutional representation.** Recurrent representations of SSM are not practical in training because explicit calculation of hidden states for every time step requires $\mathcal{O}(N^2 L)$ in time and $\mathcal{O}(NL)$ in space for a sequence of length $L$[1]. This materialization of hidden states significantly restricts RNN-based methods in scaling to long sequences. To efficiently train an SSM, the recurrence equation can be fully unrolled, assuming zero initial hidden states, as

$$
\begin{aligned}
\mathbf{h}_{t_0} &= \bar{\boldsymbol{B}} x_{t_0} & \mathbf{h}_{t_1} &= \bar{\boldsymbol{A}}\bar{\boldsymbol{B}} x_{t_1} + \bar{\boldsymbol{B}} x_{t_0} & \mathbf{h}_{t_2} &= \bar{\boldsymbol{A}}^2 \bar{\boldsymbol{B}} x_{t_2} + \bar{\boldsymbol{A}}\bar{\boldsymbol{B}} x_{t_1} + \bar{\boldsymbol{B}} x_{t_0} & &\ldots \\
y_{t_0} &= \boldsymbol{C}\bar{\boldsymbol{B}} x_{t_0} & y_{t_1} &= \boldsymbol{C}\bar{\boldsymbol{A}}\bar{\boldsymbol{B}} x_{t_1} + \boldsymbol{C}\bar{\boldsymbol{B}} x_{t_0} & y_{t_2} &= \boldsymbol{C}\bar{\boldsymbol{A}}^2 \bar{\boldsymbol{B}} x_{t_2} + \boldsymbol{C}\bar{\boldsymbol{A}}\bar{\boldsymbol{B}} x_{t_1} + \boldsymbol{C}\bar{\boldsymbol{B}} x_{t_0} & &\ldots
\end{aligned}
$$

and more generally as,

$$
y_{t_k} = \boldsymbol{C}\bar{\boldsymbol{A}}^k \bar{\boldsymbol{B}} x_{t_k} + \boldsymbol{C}\bar{\boldsymbol{A}}^{k-1}\bar{\boldsymbol{B}} x_{k-1} + \cdots + \boldsymbol{C}\bar{\boldsymbol{B}} x_{t_0}
$$

For an input sequence $x = (x_{t_0}, x_{t_1}, \ldots, x_{t_L})$, one can observe that the output sequence $y = (y_{t_0}, y_{t_1}, \ldots, y_{t_L})$ can be computed using a convolution with a skip connection

$$
y = \boldsymbol{C}\boldsymbol{K} * x + \boldsymbol{D}x, \quad \text{where } \boldsymbol{K} = (\bar{\boldsymbol{B}}, \bar{\boldsymbol{A}}\bar{\boldsymbol{B}}, \ldots, \bar{\boldsymbol{A}}^{L-1}\bar{\boldsymbol{B}}, \bar{\boldsymbol{A}}^L \bar{\boldsymbol{B}})
\tag{3}
$$

This is the well-known connection between SSM and convolution (Oppenheim & Schafer, 1975; Chen, 1984; Chilkuri & Eliasmith, 2021; Romero et al., 2021; Gu et al., 2020; 2021; 2022) and it can be computed very efficiently with a Fast Fourier Transform (FFT), which scales better than explicit matrix multiplication at each step.

## 3.2 VARIATIONAL AUTOENCODER (VAE)

VAEs (Kingma & Welling, 2013; Burda et al., 2015) are a highly successful paradigm in learning latent representations of high dimensional data and is remarkably capable at modeling complex distributions. A VAE introduces a joint probability distribution between a latent variable $\mathbf{z}$ and an observed random variable $\mathbf{x}$ of the form

$$
p_\theta(\mathbf{x}, \mathbf{z}) = p_\theta(\mathbf{x} \mid \mathbf{z})p(\mathbf{z})
$$

where $\theta$ represents learnable parameters.

The prior $p(\mathbf{z})$ over the latent is usually chosen as a standard Gaussian distribution, and the conditional distribution $p_\theta(\mathbf{x} \mid \mathbf{z})$ is defined through a flexible non-linear mapping (such as a neural network) taking $\mathbf{z}$ as input. Such highly flexible non-linear mappings often lead to an intractable posterior $p_\theta(\mathbf{z} \mid \mathbf{x})$. Therefore, an inference model with parameters $\phi$ parametrizing $q_\phi(\mathbf{z} \mid \mathbf{x})$ is introduced as an approximation which allows learning through a variational lower bound of the marginal likelihood:

$$
\log p_\theta(\mathbf{x}) \geq -D_{\mathrm{KL}}(q_\phi(\mathbf{z} \mid \mathbf{x}) \| p(\mathbf{z})) + \mathbb{E}_{q_\phi(\mathbf{z}|\mathbf{x})}\left[\log p_\theta(\mathbf{x} \mid \mathbf{z})\right]
\tag{4}
$$

where $D_{\mathrm{KL}}(\cdot \| \cdot)$ is the Kullback-Leibler divergence between two distributions.

---

[1]Further explanations in Appendix A.1

**VAE for sequences.** Sequence data can be modeled in many different ways since the latent space can be chosen to encode information at different levels of granularity, *i.e.* $\mathbf{z}$ can be a single variable encoding entire trajectories or a sequence of variables of the same length as the trajectories. We focus on the latter.

Given observed sequence variables $\mathbf{x}_{\leq T}$ up to time $T$ discretized into sequence $(\mathbf{x}_{t_0}, \ldots, \mathbf{x}_{t_{L-1}})$ of length $L$ where $t_{L-1} = T$, a sequence VAE model with parameters $\theta, \lambda, \phi$ learns a generative and inference distribution

$$p_{\theta,\lambda}(\mathbf{x}_{\leq t_{L-1}}, \mathbf{z}_{\leq t_{L-1}}) = \prod_{i=0}^{L-1} p_\theta(\mathbf{x}_{t_i} \mid \mathbf{x}_{<t_i}, \mathbf{z}_{\leq t_i}) p_\lambda(\mathbf{z}_t \mid \mathbf{z}_{<t_i})$$

$$q_\phi(\mathbf{z}_{\leq t_{L-1}} \mid \mathbf{x}_{\leq t_{L-1}}) = \prod_{i=0}^{L-1} q_\phi(\mathbf{z}_{t_i} \mid \mathbf{x}_{\leq t_i})$$

where $\mathbf{z}_{\leq t_{L-1}} = (\mathbf{z}_{t_0}, \ldots, \mathbf{z}_{t_{L-1}})$ is the corresponding latent variable sequence. The approximate posterior $q_\phi$ is explicitly factorized as a product of marginals due to efficiency reasons we shall discuss in the next section. Given this form of factorization, the variational lowerbound has been considered for discrete sequence data (Chung et al., 2015) with the objective

$$\mathbb{E}_{q_\phi(\mathbf{z}_{\leq t_{L-1}} \mid \mathbf{x}_{\leq t_{L-1}})} \left[ -\sum_{i=0}^{L-1} D_{\mathrm{KL}}(q_\phi(\mathbf{z}_{t_i} \mid \mathbf{x}_{\leq t_i}) \| p_\lambda(\mathbf{z}_i \mid \mathbf{z}_{<t_i})) + \log p_\theta(\mathbf{x}_{t_i} \mid \mathbf{x}_{<t_i} \mathbf{z}_{\leq t_i}) \right] \quad (5)$$

## 4 METHOD

In this section, we introduce *Latent S4* (LS4), a latent variable generative model parameterized using SSMs. We show how SSMs can parametrize the generative distribution $p_\theta(\mathbf{x}_{\leq T} | \mathbf{z}_{\leq T}) p_\lambda(\mathbf{z}_{\leq T})$, the prior distribution $p_\lambda(\mathbf{z}_{\leq T})$ and the inference distribution $q_\phi(\mathbf{z}_{\leq T} \mid \mathbf{x}_{\leq T})$ effectively. For the purpose of exposition, we can assume $z_t, x_t$ are scalars at any time step $t$. Their generalization to arbitrary dimensions is discussed in Section 4.4.

We first define a structured state space model with two input streams and use this as a building block for our generative model. It is an SSM of the form

$$\begin{aligned} \frac{d}{dt}\mathbf{h}_t &= \boldsymbol{A}\mathbf{h}_t + \boldsymbol{B}x_t + \boldsymbol{E}z_t \\ y_t &= \boldsymbol{C}\mathbf{h}_t + \boldsymbol{D}x_t + \boldsymbol{F}z_t \end{aligned} \quad (6)$$

where $x, y, z \in \mathcal{C}([0,T], \mathbb{R})$ are continuous real signals on time interval $[0,T]$. We denote $H(x, z, \boldsymbol{A}, \boldsymbol{B}, \boldsymbol{E}, \mathbf{h}_0, t) = H_\beta(x, z, \mathbf{h}_0, t)$, where $\beta$ denotes trainable parameters $\boldsymbol{A}, \boldsymbol{B}, \boldsymbol{E}$, as the deterministic function mapping from signals $x, z$ to $\mathbf{h}_t$, the state at time $t$, given initial state $\mathbf{h}_0$ at time 0. The above SSM can be compactly represented by

$$y_t = \boldsymbol{C}H_\beta(x, z, \mathbf{h}_0, t) + \boldsymbol{D}x_t + \boldsymbol{F}z_t \quad (7)$$

When the continuous-time input signals are discretized into discrete-time sequences $(x_{t_0}, \ldots, x_{t_{L-1}})$ and $(z_{t_0}, \ldots, z_{t_L})$, the corresponding hidden state at time $t_k$ has a convolutional view (assuming $\boldsymbol{D} = \boldsymbol{F} = \boldsymbol{0}$ for simplicity)

$$y_{t_k} = \boldsymbol{C}\boldsymbol{K}_{t_k} * x_{t_k} + \boldsymbol{C}\hat{\boldsymbol{K}}_{t_k} * z_{t_k}, \text{ where } \boldsymbol{K}_{t_k} = \bar{\boldsymbol{A}}^k\bar{\boldsymbol{B}}, \ \hat{\boldsymbol{K}}_{t_k} = \bar{\boldsymbol{A}}^k\bar{\boldsymbol{E}} \quad (8)$$

which can be evaluated efficiently using FFT. Additionally, $\boldsymbol{A}$ is HiPPO-initialized (Gu et al., 2021) for all such SSM blocks.

### 4.1 LATENT SPACE AS STRUCTURED STATE SPACE

The goal of the prior model is to realize a sequence of random variables $(z_{t_0}, z_{t_1}, \ldots, z_{t_L})$, which the prior distribution $p_\lambda(z_{\leq t_L})$ models autoregressively. Suppose $(z_{t_0}, z_{t_1}, \ldots, z_{t_n})$ is a sequence of latent variables up to time $t_n$, we define the prior distribution of $z_{t_n}$ autoregressively as

$$p_\lambda(z_{t_n} \mid z_{<t_n}) = \mathcal{N}(\mu_{z,n}(z_{<t_n}, \lambda), \sigma_{z,n}^2(z_{<t_n}, \lambda)) \quad (9)$$

where the mean $\mu_{z,n}$ and standard deviation $\sigma_{z,n}$ are deterministic functions of previously generated $z_{<t_n}$ parameterized by $\lambda$. To parameterize the above distribution, we first define an intermediate building block, a stack of which will produce the wanted distribution.

**LS4 prior block.** The forward pass through our SSM is a two–step procedure: first, we consider the latent dynamics of $z$ on $[t_0, t_{n-1}]$ where we simply leverage Equation 6 to define the hidden states to follow $H_{\beta_1}(0, z, 0, t)$. Second, on $(t_{n-1}, t_n]$, since no additional $z$ is available in this interval, we ignore additional input signals in the ODE and only include the last given latent, *i.e.* $z_{t_{n-1}}$, as an auxiliary signal for the outputs, which can be compactly denoted, with a final GELU non-linearity, as

$$y_{z,n} = \text{GELU}(\boldsymbol{C}_{y_z} H_{\beta_1}(0, 0, \underbrace{H_{\beta_2}(0, z_{[t_0, t_{n-1}]}, h_{t_{n-1}}, \boldsymbol{0}, t_{n-1})}, t_n) + \boldsymbol{F}_{y_z} z_{t_{n-1}}) \tag{10}$$

We call this *LS4 prior layer*, which we use to build our *LS4 prior block* that is built upon a ResNet structure with a skip connection, denoted as

$$\text{LS4}_{\text{prior}}(z_{[t_0, t_{n-1}]}, \psi) = \text{LayerNorm}(\boldsymbol{G}_{y_z} y_{z,n} + b_{y_z}) + z_{t_{n-1}} \tag{11}$$

where $\psi$ denotes the union of parameters $\beta_i, \boldsymbol{C}_{y_z}, \boldsymbol{F}_{y_z}, \boldsymbol{G}_{y_z}, b_{y_z}$. We define the final parameters $\mu_{z,n}$ and $\sigma_{z,n}$ for the conditional distribution in the autoregressive model as the result of a stack of *LS4 prior blocks*. During generation, as an initial condition, $z_{t_0} \sim \mathcal{N}(\mu_{z,0}, \sigma_{z,0}^2)$ where $\mu_{z,0}, \sigma_{z,0}$ are learnable parameters, and subsequent latent variables are generated autoregressively. We specify our architecture in Appendix C and use $\lambda$ to denote the union of all trainable parameters.

## 4.2 GENERATIVE MODEL

Given the latent variables, we now specify a decoder that represents the distribution $p_\theta(x_{\leq t_L} | z_{\leq t_L})$. Suppose $z_{\leq t_L}$ is a latent path generated via the latent state space model, the output path $x_{\leq t_L}$ also follows the state space formulation. Assuming we have generated $(x_{t_0}, \ldots, x_{t_{n-1}})$ and $(z_{t_0}, \ldots, z_{t_n})$, the conditional distribution of $x_{t_n}$ is parametrized as

$$p_\theta(x_{t_n} | x_{<t_n}, z_{\leq t_n}) = \mathcal{N}(\mu_{x,n}(x_{<t_n}, z_{\leq t_n}, \theta), \sigma_x^2) \tag{12}$$

where $\sigma_x$ is a pre-defined observation standard deviation and $\mu_{x,n}$ is a deterministic function of $z_{\leq t_n}$ and $x_{<x_n}$.

**LS4 generative block.** Different from the prior block, both observation and latent sequences are input into our model, and we define intermediate outputs $g_{x,n}$ and $g_{z,n}$ as

$$\begin{aligned}
h_{t_n} &= H_{\beta_3}(0, z_{t_{n-1}}, H_{\beta_4}(x_{[t_0, t_{n-1}]}, z_{[t_0, t_{n-1}]}, \boldsymbol{0}, t_{n-1}), t_n) \\
g_{x,n} &= \text{GELU}(\boldsymbol{C}_{g_x} h_{t_n} + \boldsymbol{D}_{g_x} x_{t_{n-1}} + \boldsymbol{F}_{g_x} z_{t_n}) \\
g_{z,n} &= \text{GELU}(\boldsymbol{C}_{g_z} h_{t_n} + \boldsymbol{D}_{g_z} x_{t_{n-1}} + \boldsymbol{F}_{g_z} z_{t_n})
\end{aligned} \tag{13}$$

which are used to build a *LS4 generative block* defined as

$$\begin{aligned}
\hat{g}_{x,n} &= \text{LayerNorm}(\boldsymbol{G}_{g_x} g_{x,n} + b_{g_x}) + x_{t_{n-1}} \\
\hat{g}_{z,n} &= \text{LayerNorm}(\boldsymbol{G}_{g_z} g_{z,n} + b_{g_z}) + z_{t_n} \\
\text{LS4}_{\text{gen}}(x_{[t_0, t_{n-1}]}, z_{[t_0, t_n]}, \psi) &= (\hat{g}_{x,n}, \hat{g}_{z,n})
\end{aligned} \tag{14}$$

where $\psi$ denotes all parameters inside the block. Note that the generative block gives two streams of outputs, which can be used as inputs for the next stack. We then define the final mean value $\mu_{x,n}$ as the result of a stack of *LS4 generative blocks*. The initial condition for generation is given as $x_{t_0} \sim \mathcal{N}(\mu_{x,0}(z_0, \theta), \sigma_x)$ where $\mu_{x,0}$ exactly follows our formulation while taking only $z_{t_0}$ as input. The subsequent $x_{t_n}$'s are generated autoregressively. We specify our architecture in Appendix C and use $\theta$ to denote the union of all trainable parameters.

## 4.3 INFERENCE MODEL

The latent variable model up to time $t_n$ has intractable posterior $p_\theta(z_{\leq t_n} \mid x_{\leq t_n})$. Therefore, we approximate this distribution with $q_\phi(z_{\leq t_n} \mid x_{\leq t_n})$ using variational inference.

We parameterize the inference distribution at time $t_n$ to depend only on the observed path $x_{\leq t_n}$:

$$q_\phi(z_t \mid x_{\leq t_n}) = \mathcal{N}(\hat{\mu}_{z,t_n}(x_{\leq t_n}, \phi), \hat{\sigma}_{z,t_n}^2(x_{\leq t_n}, \phi)) \tag{15}$$

**LS4 inference block.** The inference block is defined as

$$\hat{y}_{z,n} = \text{GELU}(\boldsymbol{C}_{\hat{y}_z} H_{\beta_5}(x_{[t_0,t_n]}, 0, \boldsymbol{0}, t_{n-1}) + \boldsymbol{D}_{\hat{y}_z} x_{t_n})$$

$$\text{LS4}_{\text{inf}}(x_{[t_0,t_n]}, \psi) = \text{LayerNorm}(\boldsymbol{G}_{\hat{y}_z} \hat{y}_{z,n} + b_{\hat{y}_z}) + x_{t_n} \tag{16}$$

Notice that input $x$ is fully present in $[t_0, t_n]$ unlike in the generative model. $\hat{\mu}_{z,t}$ and $\hat{\sigma}_{z,t}$ are each a deep stack of our inference blocks similarly defined as before. Here we also justify our choice of factorizing posterior as a product of marginals as in Equation 5. By having each $z_t$ explicitly depending on $x_{\leq t_n}$ only, we can leverage the fast convolution operation to obtain all $z_t$ in parallel, thus achieving fast inference time, unlike the autoregressive nature of the prior and generative model.

## 4.4 LS4: PROPERTIES AND PRACTICE

We highlight some properties of LS4. In particular, we compare in the following proposition the expressive power of our generative model against structured state space models.

**Proposition 4.1.** *(LS4 subsumes S4.) Given any autoregressive model $r(x)$ with conditionals $r(x_n|x_{<n})$ parameterized via deep S4 models, there exists a choice of $\theta, \lambda, \phi$ such that $p_{\theta,\lambda}(x) = r(x)$ and $p_{\theta,\lambda}(z|x) = q_\phi(z|x)$, i.e. the variational lower bound (ELBO) is tight.*

A proof sketch is provided in Appendix B. This result shows that LS4 subsumes autoregressive generative models based on vanilla S4 (Gu et al., 2021), given that the architecture between SSM layers is the same. Crucially, with the assumption that we are able to globally optimize the ELBO training objective, LS4 will fit the data at least as well as vanilla S4.

**Scaling to arbitrary feature dimensions.** So far we have assumed the input and latent signals are real numbers. The approach can be scaled to arbitrary dimensions of inputs and latents by constructing LS4 layers for each dimension which are input into a mixing linear layer. We call such parallelized SSMs *heads* and provide a pseudo-code in Appendix C.

**Proposition 4.2.** *(Efficiency.) For a SSM with $H$ heads, an observation sequence of length $L$ and hidden dimension $N$ can be calculated in $\mathcal{O}(H(L+N)\log(L+N))$ time and $\mathcal{O}(H(L+N))$ space.*

We provide proof in Appendix B. Note that our model is much more efficient in both time and space than RNN/ODE-based methods (which requires $\mathcal{O}(N^2 L)$ in time and $\mathcal{O}(NL)$ in space as discussed in Section 3.1). To demonstrate the computation efficiency, we additionally provide below pseudo-code for a single LS4 prior layer 10. The other blocks can be similarly constructed.

```
def LS4_prior_layer(z,  A, B, C, F, h_0): # z: (B, L, 1)
    K = C @ materialize_kernel(z, A, B, h_0) #  O((L+N)log(L+N)) time
    CH = fft_conv(K, z) # O(LlogL) time and O(L) space
    y = gelu(CH + F * z)
    return y
```

Note that in practice, $\boldsymbol{A}$ is HiPPO initialized (Gu et al., 2020) and the materialized kernel includes $\boldsymbol{C}$ so that the convolution is computed directly in the projected space, bypassing materializing the high-dimensional hidden states.

## 5 EXPERIMENTS

In this section, we verify the modeling capability of LS4 empirically. There are three main questions we seek to answer: (1) How effective is LS4 in modeling stiff sequence data? (2) How expressive is LS4 in scaling to real time-series with a variety of temporal dynamics? (3) How efficient in training and inference is LS4 in terms of wall-clock time?

### 5.1 LEARNING TO GENERATE DATA FROM STIFF SYSTEMS

Modeling data generated by dynamics with widely separated time scales has been proven to be particularly challenging for vanilla ODE-based approaches which make use of standard explicit solvers for inference and gradient calculation. Kim et al. (2021) showed that as the learned dynamics *stiffen* up to track data paths, the ODE numerics start to catastrophically fail; the inference cost

raises drastically and the gradient estimation process becomes ill–conditioned. These issues can be mitigated by employing implicit ODE solvers or *ad-hoc* rescalings of the learned vector field (see (Kim et al., 2021) for further details).

In turn, state–space models do not suffer from stiffness of dynamics as the numerical methods are sidestepped in favor of an exact evaluation of the convolution operator. We hereby show that LS4 is able to model data generated by a prototype stiff system.

**FLAME problem** We consider a simple model of flame growth (FLAME) (Wanner & Hairer, 1996), which has been extensively studied as a representative of highly stiff systems:

$$\frac{\mathrm{d}}{\mathrm{d}t} x_t = x_t^2 - x_t^p$$

where $p \in \{3, 4, \ldots, 10\}$. For each $p$, 1000 trajectories are generated for $t \in [0, 1000]$ with unit increment.

**Generation.** In Figure 1a, we show the mean trajectories and the distribution at each time step and that our samples closely match the ground-truth data. The Latent ODE (Rubanova et al., 2019) instead fails to do so and produces non–stiff samples drastically different from the target.

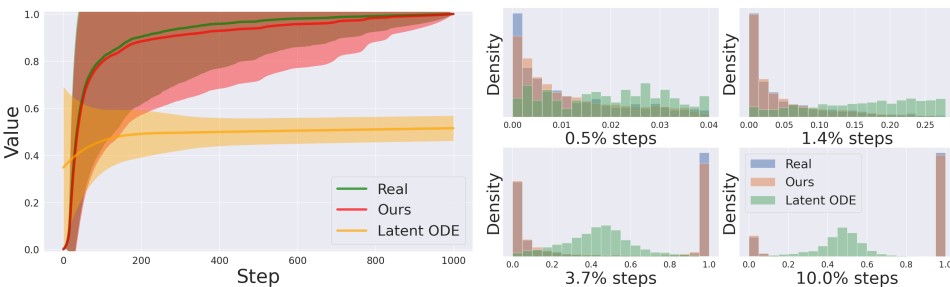

(a) Generation of the stiff system.

(b) Marginal histograms at steps equally spaced between the 0.5% and 10% steps in log scale.

**Marginal Distribution.** We plot the marginal distribution of the real data and the generated data from both our model and Latent ODE. Since the stiff transitions are mostly distributed before 10% of total steps, we visualize the marginal histograms at 4 time steps equally spaced between the 0.5% and 10% steps in log scale (see Figure 1b). We observe that the empirical histogram matches the ground truth distribution significantly better than what is produced by Latent ODEs, as also qualitatively visible from the samples in (a).

## 5.2 GENERATION WITH REAL TIME-SERIES DATASETS

We investigate the generative capability of LS4 on real time-series data. We show that our model is capable of fitting a wide variety of time-series data with significantly different temporal dynamics. We leave implementation details to Appendix D.1.

**Data.** We use Monash Time Series Repository (Godahewa et al., 2021), a comprehensive benchmark containing 30 time-series datasets collected in the real world, and we choose FRED-MD, NN5 Daily, Temperature Rain, and Solar Weekly as our target datasets. Each dataset exhibits unique temporal dynamics, which makes generative learning a challenging task. A sample from each dataset can be visualized in Figure 2.

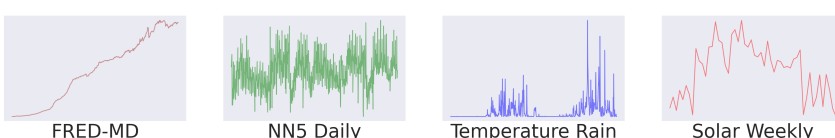

Figure 2: Monash data. The selected datasets exhibit a variety of temporal dynamics ranging from relatively smooth to stiff transitions.

**Metrics.** We propose 3 different metrics for measuring generation performance, namely *Marginal*, *Classification*, and *Prediction* scores. *Marginal* scores calculate the absolute difference between

empirical probability density functions of two distributions – the lower the better (Ni et al., 2020). Following Kidger et al. (2021), we define *Classification* scores as using a sequence model to classify whether a sample is real or generated and use its cross-entropy loss as a proxy for generation quality – the higher the less distinguishable the samples, thus better the generation. Similarly, *Prediction* scores use a train-on-synthetic-test-on-real seq2seq model to predict $k$ steps into the future – the lower the more predictable, thus better the generation. We use a 1-layer S4 (Gu et al., 2021) for both Classification and Prediction scores (see Appendix D.1 for more details). We will discuss the necessity of all 3 metrics in the following discussion section.

We compare our model with several time-series generative models, namely the VAE-based methods such as RNN-VAE (Rubanova et al., 2019), GP-VAE (Fortuin et al., 2020), ODE$^2$VAE (Yildiz et al., 2019), Latent ODE (Rubanova et al., 2019), GAN-based methods such as TimeGAN (Yoon et al., 2019) and SDE GAN (Kidger et al., 2021), and SaShiMi (Goel et al., 2022). We show quantitative results in Table 1.

| Data | Metric | RNN-VAE | GP-VAE | ODE$^2$VAE | Latent ODE | TimeGAN | SDEGAN | SaShiMi | **LS4 (Ours)** |
|---|---|---|---|---|---|---|---|---|---|
| FRED-MD | Marginal ↓ | 0.132 | 0.152 | 0.122 | 0.0416 | 0.0813 | 0.0841 | 0.0482 | **0.0221** |
| | Class. ↑ | 0.0362 | 0.0158 | 0.0282 | 0.327 | 0.0294 | 0.501 | 0.00119 | **0.544** |
| | Prediction ↓ | 1.47 | 2.05 | 0.567 | **0.0132** | 0.0575 | 0.677 | 0.232 | 0.0373 |
| NN5 Daily | Marginal ↓ | 0.137 | 0.117 | 0.211 | 0.107 | 0.0396 | 0.0852 | 0.0199 | **0.00671** |
| | Class. ↑ | 0.000339 | 0.00246 | 0.00102 | 0.000381 | 0.00160 | 0.0852 | 0.0446 | **0.636** |
| | Prediction ↓ | 0.967 | 1.169 | 1.19 | 1.04 | 1.34 | 1.01 | 0.849 | **0.241** |
| Temp Rain | Marginal ↓ | 0.0174 | 0.183 | 1.831 | **0.0106** | 0.498 | 0.990 | 0.758 | 0.0834 |
| | Class. ↑ | 0.00000212 | 0.0000123 | 0.0000319 | 0.0000419 | 0.00271 | 0.0169 | 0.0000167 | **0.976** |
| | Prediction ↓ | 159 | 2.305 | 1.133 | 145 | 1.96 | 2.46 | 2.12 | **0.521** |
| Solar Weekly | Marginal ↓ | 0.0903 | 0.308 | 0.153 | 0.0853 | 0.0496 | 0.147 | 0.173 | **0.0459** |
| | Class. ↑ | 0.0524 | 0.000731 | 0.0998 | 0.0521 | 0.6489 | 0.591 | 0.00102 | **0.683** |
| | Prediction ↓ | 1.25 | 1.47 | 0.761 | 0.973 | 0.237 | 0.976 | 0.578 | **0.141** |

Table 1: Generation results on FRED-MD, NN5 Daily, Temperature Rain, and Solar Weekly.

Our model significantly outperforms the baselines on all datasets. We note that baseline models have a hard time modeling NN5 Daily and Temperature Rain where the transition dynamics are stiff. For Temperature Rain where most data points lie around $x$-axis with sharp spikes throughout, latent ODE generates mostly closely to the $x$-axis, thus achieving lower marginal scores, but its generation is easily distinguishable from data, thus making it a less favorable generative model. We demonstrate that Marginal scores alone are an insufficient metric for generation quality. SaShiMi, an autoregressive model based on S4, does not perform as well on time series generation in the tasks considered. We further discuss the reason in Appendix D.1.

## 5.3 INTERPOLATION & EXTRAPOLATION

We also show that our model is expressive enough to fit to irregularly-sampled data and performs favorably in terms of interpolation and extrapolation. Interpolation refers to the task of predicting missing data given a subset of a sequence while extrapolation refers to the task that data is separated into 2 segments each with half the length of the full sequence, and one predicts the latter segment given the former.

**Data.** Following Rubanova et al. (2019); Schirmer et al. (2022), we use USHCN and Physionet as our datasets of choice. The United States Historical Climatology Network (USHCN) (Menne et al., 2015) is a climate dataset containing daily measurements form 1,218 weather stations across the US for precipitation, snowfall, snow depth, minimum and maximum temperature. Physionet (Silva et al., 2012) is a dataset containing health measurements of 41 signals from 8,000 ICU patients in their first 48 hours. We follow preprocessing steps of Schirmer et al. (2022) for training and testing.

**Metrics.** We use mean squared error (MSE) to evaluate both interpolation and extrapolation.

We compare our model with RNN (Rumelhart et al., 1985), RNN-VAE (Chung et al., 2014; Rubanova et al., 2019), ODE-RNN (Rubanova et al., 2019), GRU-D (Rubanova et al., 2019), Latent ODE (Chen et al., 2018; Rubanova et al., 2019), and CRU Schirmer et al. (2022). Results are shown in Table 2.

---

[2] Numbers are taken from the original paper. We keep the significant digits unchanged

| Task | Data | RNN | RNN-VAE | ODE-RNN | GRU-D | Latent ODE | CRU [2] | LS4 (Ours) |
|---|---|---|---|---|---|---|---|---|
| Interp. | Physionet | 2.918 | 5.930 | 2.234 | 3.325 | 8.341 | 1.82 | **0.6287** |
| | USHCN | 4.322 | 7.561 | 2.473 | 3.395 | 6.859 | 0.16 | **0.0594** |
| Extrap. | Physionet | 3.406 | 3.064 | **3.014** | 3.120 | 4.212 | 6.29 | 4.942 |
| | USHCN | 9.474 | 9.083 | 9.045 | 8.964 | 8.959 | 12.73 | **2.975** |

Table 2: Interpolation and extrapolation MSE ($\times 10^{-3}$) scores. Lower scores are better.

We observe that our model outperforms previous continuous-time methods. Our model performs less well on extrapolation for Physionet compared to ODE-RNN and latent ODE. We postulate that this is due to the high variability granted by our latent space. Since new latent variables are generated as we extrapolate, our model generates paths that are more flexible (hence less predictable) than those of Latent ODE, which instead uses a single latent variable to encode an entire path. We additional observe that our model achieves ELBO of $-669.0$ and $-250.2$ on Physionet interpolation and extrapolation tasks respectively. These bounds are much tighter lower bounds than other VAE-based methods, *i.e.* RNN-VAE, which reports $-412.8$ and $-220.2$, and latent ODE, which reports $-410.3$ and $-168.5$. We leave additional ELBO comparisons in Appendix D.

## 5.4 RUNTIME

We additionally verify the computational efficiency of our model for both training and inference. We do so by training and inferring on synthetic data with controlled lengths specified below.

**Data.** We create a set of synthetic datasets with lengths $\{80, 320, 1280, 5120, 20480\}$ to investigate scaling of training/inference time with respect to sequence length. Training is done with 100 iterations through the synthetic data, and inference is performed given one batch of synthetic data (see Appendix D.3).

**Metrics.** We use wall-clock runtime measured in milliseconds.

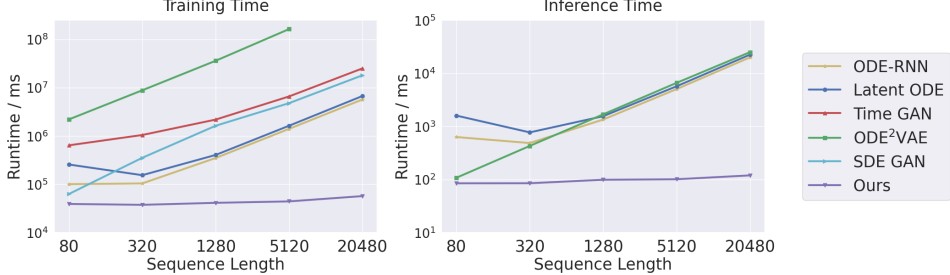

Figure 3: Runtime comparison. The $y$-axis shows run-time (**ms**) of each setting in log scale. Our runtime stays consistently lower across all sequence lengths investigated.

Figure 3 shows model runtime in log scale. ODE$^2$VAE fails to finish training on the last sequence length within a reasonable time frame, so we omit its plot of the last data point. Our model performs consistently and significantly lower than baselines, which are observed to scale linearly with input lengths, and is $\times 100$ faster than baselines in both training and inference on 20480 length.

## 6 CONCLUSION

We introduce LS4, a powerful generative model with latent space evolution following a state space ODE. Our model is built with a deep stack of LS4 prior/generative/inference blocks, which are trained via standard sequence VAE objectives. We also show that under specific choices of model parameters, LS4 subsumes autoregressive S4 models. Experimentally, we demonstrate the modeling power of LS4 on datasets with a wide variety of temporal dynamics and show significant improvement in generation/interpolation/extrapolation quality. In addition, our model shows $\times 100$ speed-up in training and inference time on long sequences. LS4 demonstrates improved expressivity and computational efficiency, and we believe that it has a further role to play in modeling general time-series sequences.

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
