# OpenReview forum: "Deep Latent State Space Models for Time-Series Generation"
_ICLR.cc/2023/Conference — Submitted to ICLR 2023_

### Official Review · Reviewer_PiDm · 2022-10-24

**Confidence:** 3
**Correctness:** 3
**Technical Novelty And Significance:** 2
**Empirical Novelty And Significance:** 3
**Recommendation:** 5

**Clarity, Quality, Novelty And Reproducibility:**

The paper is clearly written in terms of language. But I think there are several gaps that need to be filled. Reading the paper now a few times still leaves me confused about several details. Examples are given in Strengths and Weaknesses. I think these need to be improved, but many of them can be addressed easily.

I think the paper is not particularly novel, as it is a direct extension of previous work (S4), combining it with variational inference.
This combination did not require novel technical contributions. On the other hand, this is the first paper proposing a probabilistic extension of S4.

In the current form, I would not be able to reproduce the paper, but I hope to gain more clarity from the rebuttal/revision.

**Strength And Weaknesses:**

While the paper is generally well written, clarification is needed for several details:

For instance, the model and posterior before Eq. (5) makes assumptions that need explanation:
- The emission model is both auto-regressive in all observations x and all latent variables z. Can you motivate why you consider the dependence structure on x additionally to z? And why use auto-regressive instead Markovian models which are more common in SSM literature?
- The inference model assumes that the (approximate) posterior factorises as a product of marginals, which is very restrictive. Why is this choice made as a starting point?
- Eq. (5) appears to be in conflict with the equation for the approximate posterior above. Whereas above, you assume that the approximate posterior factorises as a product, here you use a factorisation into conditionals (which is not restrictive). Which factorisation did you actually consider? And is Eq. (5) the objective you use in the end? It's not obvious as that is in the background section.
- In Eq (5), the posterior is conditioned on observation only until some intermediate x_ti, but it should be the full trajectory of observations, otherwise you have an additional gap in the ELBO. This has been discussed e.g. in [1]. (Unless you would do a filtering-based approximation for all the conditionals p(y_t | y_<t), but that seems also not the case.)

Another thing that confused me is that in the beginning x is used as inputs and y as target observations. And later, x are the observations of the generative model. I think it would be good to either use a different symbol for one, or clarify this better.

I don’t understand Eq. (8). x_tk and z_tk are scalars, and h_tk is the feature vector at time-step t. How does the convolution operation apply here? Maybe you want to write the convolution for the sequence, similar to Eq. (3)?

What is H_\beta? In the inline text above its mentioned that it is the function mapping x, z to h. What is that function? Only dh/dt has been specified, but not h itself, or how it is approximated. Bilinear transformation?
Later, there is H_\beta_1 and other subscripts. This is undefined. What is the index?

In 4.1, before starting the paragraph "LS4 prior block", I am missing a connecting sentence. It took me a while to understand that the block parametrises the conditional distribution. Because the next equation has "y" as output and there is no other equation for z or mu_z and sigma_z, this paragraph read as unconnected to that equation. This can be streamlined better for the reader.

I am also a bit worried and uncertain in my understanding about how a latent trajectory is generated. For the posterior, as far as I understand, you can compute the approximate posterior marginals in parallel, because you factorise it as a product of marginals? If so, this should be noted. For the prior, can you only compute the hiddens h and y in parallel, and then have to sample auto-regressively? Is this the reason why there are no forecasting experiments? If this is a limitation, it would be great to state this explicitly.

Why is the prior on z discrete-time, whereas h is continuous-time? I think this is never mentioned.

Eq (11), what is G_yz?  suppose its just an additional matrix. Why this additional transformation here?

“In turn, state–space models do not suffer from stiffness of dynamics as the numerical methods are sidestepped in favor of an exact evaluation of the convolution operator.”
I would like to learn a bit more about this. As far as I understand, the convolution operator is applied to the *discretized* system. How does this sidestep numerical problems?

How do you deal with initialisation? From what I understand from the previous work, the Hippo initialisation is one of the key things for the empirical performance. I think this was not discussed, despite its importance. And there is also no theory regarding how to adjust this for the additional latent variables z.

Related work.
I think the paper is missing a quite large literature of related work on deep state-space models that are not of the S4 type. Among the vast literature on deep variational state space models, several methods even use conditional linear or linearised state space models, for instance [2,3,4,5].


Experimental evaluation.
The performed experiments show that the model is able to generate realistic samples even for quite challenging data distributions. This seems quite promising. However, I find the choice of experiments and baselines not ideal. In particular,
- why not compare to the obvious baseline S4? Both in terms of speed and performance.
- since the model uses latent variables to express uncertainty, I would have expected to see some experiments and evaluation metrics that would benefit from this, e.g. probabilistic forecasting evaluated with the CRPS score.
The current set of experiments leave me somewhat undecided. On the one hand, the model seems to be able to generate realistic data very well. I am not sure if this implies that the latent transition model is also learned properly, or whether there is potentially a short-cut, e.g. using only the initial latent state and ignoring the dynamics. This speculation would be easily debunked through a forecasting experiment. In the experiments in 5.3, the worse performance on Physionet is explained as if the randomness in the latent space was a limitation, but that should be actually the selling point of this model, as certain data just is not perfectly predictable.


[1] Bayer et al. 2021, Mind the Gap when Conditioning Amortised Inference in Sequential Latent-Variable Models
[2] Rangapuram et al. 2018, Deep State Space Models for Time Series Forecasting
[3] Fraccaro et al., 2017, A Disentangled Recognition and Nonlinear Dynamics Model for Unsupervised Learning
[4] Kurle et al. 2020, Deep Rao-Blackwellised Particle Filters for Time Series Forecasting
[5] Klushyn et al. 2021, Latent Matters: Learning Deep State-Space Models

**Summary Of The Paper:**

This paper presents an extension of S4 layers [Gu et al. 2021] with latent variables, which are used to define a generative model.
To this end, the authors define an auto-regressive prior distribution over latent variables leveraging the S4 parametrisation. Furthermore, the approximate posterior is defined as a product of marginals, and also leverages S4 parametrisation.


**Summary Of The Review:**

I think this paper has great potential and it is overall good work. However, in the current version, I find several things unclear, and would therefore vote for rejection. Many questions and clarification could be addressed in the rebuttal though.

---

> ### Author Response · Authors · 2022-11-19
> **Thank you for your review and suggestions.**
>
> Thank you for your review and suggestions. We make clarifications for the concerns below:
>
> * _Why dependence structure on z and why autoregressive instead of Markovian?_
>
> Latent variables are used to increase the expressivity of our model because the resulting marginal over $x$ is a mixture distribution. Adding dependency structure on $z$ further improves flexibility. Intuitively, a static time-independent latent $z$ corresponds to a fixed clustering of the trajectories, while adding dynamics allows the "clustering" to change over time (as a function of $x$). This is used in a variety of sequence models, including classic ones like HMM/Kalman filters.  We use a general AR structure (as opposed to Markovian) to further improve flexibility: the model can choose to ignore earlier time steps (and thus be Markovian) but it doesn't have to. Crucially, because of our use of convolutions, a more general AR structure doesn't come at additional cost.
>
> * _Why assume inference model as a product of marginals?_
>
> Thank you for noticing this! And just as you mentioned, we factorize the posterior as a product of marginals precisely because we can compute any time $z_t$ given the history efficiently. Having the autoregressive dependency prevents this efficiency property. We have added additional explanations to the paper.
>
> * _Equation 5 is in conflict with the inference factorization we used. Is Equation 5 the objective we use?_
>
> Thank you for noticing this mistake. The posterior in Eq 5 should not have explicit dependency on the history of $z$’s to be consistent with the definition above. We have updated the paper to reflect the change. The actual objective we use is Eq 5 without the posterior explicitly depending on the history of $z$’s.
>
> * _Additional gap in ELBO._
>
> Thank you for the reference. In our model we indeed only have $z_t$ depend on the $x$’s until $t$ since the SSM we consider is causal, and as mentioned in the reference, this indeed will leave a gap in the ELBO. However, it is easy to implement a bidirectional version of the inference model such that any $z_t$ also depends on the future trajectory. Such a design is mentioned in SaShiMi [1] and is beyond the scope of the models we consider in this paper.
>
> * _Confusing notation for x and y._
>
> At the beginning we are talking about general sequence to sequence mappings, where $x$ and $y$ are just general inputs and outputs. This is consistent with when we explain intermediate building blocks where the outputs are also used as $y$, which will be equivalent to either $x$ or $z$ at the final layer.
>
> * _Confusion on Equation 8 and how the convolution is computed._
>
> We have updated Eq 8 as suggested. This is closer to our actual implementation where we bypass explicitly calculating $h_t$.
>
> * _What is H_\beta and the indexed variants?_
>
> Each $h_t$ can be explicitly solved in continuous time and results in a convolution with the input $x_t$’s (as discussed in the latter section of Section 3.1). We use the same convolutional operation to compute $h_t$. When needed, we can also compute $h_t$ using fully autoregressive recurrence, as mentioned in Eq 2, and the dependence on the past input sequence is compactly represented as $H_\beta$. $\beta_1$ and other indexed $\beta$’s simply represent different sets of learnable parameters, so that it’s clear the weights are not tied.
>
> * _Missing connecting sentence._
>
> Thank you for the suggestion, we have added transition sentence for smoother reading.
>
> * _Confusion on posterior computation in parallel and autoregressive sampling._
>
> This is as explained above. We factorize posterior as a product of marginal for efficiency reasons so that we can compute using convolution. And precisely for the prior, we need to sample autoregressively when we are generating new samples. You are correct in noting that this autoregressive sampling is limitation since for generation we need to fall back to recurrent computation.
>
> * _Prior z is discrete while h is continuous?_
>
> The prior needs to be sampled autoregressively, so we need to assume a small discrete timestep to do so, otherwise $z_t$ will depend on itself rather than some z before t, if we simply reuse Eq 6. This is also why we separate the $h_t$ dynamics into $[t_0, t_{n-1}]$ and $(t_{n-1}, t_n]$, to avoid such dependence.
>
> * _Purpose of G_yz?_
>
> $G_yz$ is an additional matrix. We find that additional affine transformation with nonlinearity after SSM output increases modeling capacity.

---

> > ### Author Response · Authors · 2022-11-19
> > **Thank you for your review (continued)**
> >
> > * _How does convolution side-step numerical problems under stiff transitions?_
> >
> > We avoid the numerical issues encountered by attempting to fit stiff trajectories with other continuous-time models such as neural ordinary differential equations trained via continuous-time adjoints. The discretization step is not an issue: it is merely a matter of choosing which points of the trajectory to supervise the model with, and we choose the same set of points for all baselines. The first ingredient to avoid numerical issues is solving the system as a convolution rather than autoregressively. The resulting reverse-mode gradients are generally well-behaved in comparison to continuous-time backsolve adjoints, which require careful choice of numerical solver for the backward system (see [2]). The second ingredient is the presence of latent variables as inputs to the system. Controlled systems are easier to optimize to track stiff trajectories (Neural CDEs are [3] are an example from the neural differential equation literature).
> >
> > * _How are things initialized?_
> >
> > We use Hippo initialization for all matrices as in previous works. We have added this detail to Section 4.4.
> >
> > * _Related works._
> >
> > We thank the reviewer for the suggestions. Of course, SSMs are a foundational modeling primitives across fields, from filtering and estimation theory to control. Our approach is contextualized with respect to the closest variants of latent and deterministic SSMs under the following criteria: (1) inference is performed via efficient convolutions and (2) scaling to deep stacks of SSM layers is verified.
> >
> > * _Comparison with S4?_
> >
> > SaShiMi is the more suitable baseline choice than S4 as it is synonymous to S4 while being architecturally close to our model. The only difference is SaShiMi discretized output for audio data, while we change it to produce a continuous output for general sequences, and so it is effectively (stacked) S4.
> >
> > * _Experiments with probabilistic forecasts and CRPS scores._
> >
> > We have calculated CRPS score as suggested on USHCN and Physionet and have added a new Table 5 in Appendix D.2. Our model outperforms baselines on all settings except extrapolation on Physionet. The CRPS scores seem to be correlated with MSE mainly because the output probability distribution is fixed to be a Gaussian with mean as the model output and a predetermined std. As CRPS is calculated as the area under CDF up to the observed data point, the fixed std will not play a major factor in computing the score since only when the mean is close to the observation will the score vary. This uncertainty in the mean output likely comes from the randomness as a result of each latent variable, which we think is what makes it powerful for generating novel and diverse samples but limited in extrapolating long into the future with high accuracy given history.
> >
> > [1] Goel, Karan, et al. "It's Raw! Audio Generation with State-Space Models." arXiv preprint arXiv:2202.09729 (2022).\
> > [2] Kim, Suyong, et al. "Stiff neural ordinary differential equations." Chaos: An Interdisciplinary Journal of Nonlinear Science 31.9 (2021): 093122.\
> > [3] Kidger, Patrick, et al. "Neural controlled differential equations for irregular time series." Advances in Neural Information Processing Systems 33 (2020): 6696-6707.\

---

### Official Review · Reviewer_rvUd · 2022-10-24

**Confidence:** 4
**Correctness:** 3
**Technical Novelty And Significance:** 3
**Empirical Novelty And Significance:** Not applicable
**Recommendation:** 3

**Clarity, Quality, Novelty And Reproducibility:**

The theoretical part of the paper is clearly written but the experimental section needs clarification in order to evaluate the quality of the results.
The paper is novel in a way that it combines the S4 approach for SSM with the VAE framework for data generation.

**Strength And Weaknesses:**


 - [+] The paper is well written and easy to follow. All necessary theoretical and conceptual background information is concisely introduced.
 - [+] The reviewer also appreciates that comprehensive source code is added in the supplementary materials.
 - [-] In several passages of the text “significant” performance gains are claimed although no statistical tests are performed. Furthermore, there is no indication that several replicates per method were
       performed and there are no error bars in the respective result tables.
 - [-] The authors do not explain how the hyperparameters for the respective methods were selected. In section D only one specific hyperparameter setting of LS4 is explained: how was this selected and what
       is the exact hyperparameter search space of LS4 and the baseline methods?
 - [-] It is not explained how 4 of 30 datasets were selected in section 5.2. Why weren’t datasets from baseline methods like SaShiMi or SDEGAN selected?
 - [-] Since SaShiMi is specifically designed for modeling raw audio waveforms, wouldn’t S4 or a variant thereof have been a more suitable baseline? This is also briefly discussed by the authors in the last
       paragraph of D.1.
 - [-] The information in Figure 1 is not clearly illustrated, and the main caption is missing. The plot in (a) is quite overloaded and the y-axis label should be more specific (e.g. “value of x”). It is not
       clear how well LS4 approximates the ground truth. Is it possible to plot the mean trajectories with standard deviations for a fixed p? Some more information on the FLAME problem would be helpful. How
       do different trajectories arise given a fixed p, does the variation come from varying initial conditions?

### Questions

 - What are the runtimes of SaShiMi and other VAE-based methods in Table 1? Can the authors explain why the runtimes of the baseline methods experience exponential growth, shouldn’t this be on scale
   O(N^2 L)? Why do the runtimes of Latent ODE drop by increasing the sequence length from 80 to 320? It would also be interesting to compare memory consumption in an experiment since shorter runtime can
   come at the cost of more overhead and the practical implementation does not necessarily have to follow the theoretical result in Proposition 4.2.

**Summary Of The Paper:**

The paper proposes LS4 which is a generative model for sequences inspired by the deep state space model (SSM) S4. LS4 performs latent space evolution following a state space ODE and is trained via sequence VAE objectives. State-of-the-art results on selected datasets for continuous-time latent generative models are reported.

**Summary Of The Review:**

The paper is well written and easy to follow. Nevertheless, important information needs to be added in the description of the experimental process to be able to correctly assess the presented results.

### Minor Questions

 - Why wasn’t the FID used for comparing the generated with the real distribution in section 5.2? It is a well-established metric for generative models and e.g. the bottleneck layer of an Autoencoder could
   have been used to compute the Frechet Distance.

---

> ### Author Response · Authors · 2022-11-18
> **Thank you for your review and suggestions**
>
> Thank you for your review and suggestions! We wish to address your concerns here below:
> * _No statistical tests are performed._
>
> As suggested, we performed additional trials on the MONASH datasets. However, due to time limit we could not finish low-efficiency baselines and we only fully tested against equally fast SaShiMi baseline. 5 trials are performed and results are in a separate one-page file in supp zip. Note that all the numbers for our model reported in the main text falls within the error bar of the additional runs. The average difference in Marginal score between LS4 and SaShiMi is ~22 standard deviation averaged across all dataset, with particular gain in Temperature Rain. And the difference is ~11 and ~32 standard deviation for Classification and Prediction scores respectively averaged across all datasets.
>
> * _Lack of architectural details._
>
> For hyperparameters of LS4, we keep the hyperparameters of a single prior/generative/inference block the same as those used in [1], e.g. 64-dimensional hidden state $h_t$ and 64 parallel SSMs. We mainly experimented with the number of such blocks to stack and the dimension of latent space. We in general find 4 stacks and 5-dimensional latent space enough for modeling the data distributions. For baselines we use the default hyperparameter choices from their respective repo, and we use the same dimension of latent space as our model to keep the representational power similar. We have updated Appendix D for more detailed hyperparameter descriptions for each experiment and choices for baselines.
>
> * _Explaining how the 4 datasets from MONASH are selected. Why not baselines from S4 or SaShiMi?_
>
> SaShiMi is specifically designed for audio data with quantization and SDEGAN is designed for modeling stochastic processes. They are not designed to handle general sequences (e.g. SaShiMi can struggle on data without quantization). However, we present our model to be able to capture more complex dynamics in general sequences, agnostic to a particular type, and one type of such dynamics involves noisy stiff transitions, as is present in temperature rain data. MONASH is a comprehensive repository for such general sequences, and we choose a subset of Monash datasets to exhibit a variety of time series in terms of average 1-lag autocorrelation, which measures the 1-step correlation in time. For example, average 1-lag autocorrelation of Fred-MD , NN5 Daily and Solar weekly are 0.98, 0.38, and 0.56 respectively. This wide spectrum indicates that the selected datasets exhibit a wide variety of transition dynamics, from relatively smooth to very stiff. We show our model’s capability to successfully fit to all of them.
>
> * _Why is SaShiMi used instead of S4 as the baseline?_
>
> Sorry for the confusion. We stress that SaShiMi here is synonymous to S4 and is architecturally the closest to our model. The only difference is that it uses a quantization scheme to output discrete tokens for audio. For our baseline implementation, we only change the output to be continuous and parameterized by Gaussian distribution, same as our model, so that SaShiMi effectively becomes S4.
>
> * _Figure 1 is unclear and FLAME problem needs clarification._
>
> We have updated the plot as suggested and we hope it is more clear now. The randomness comes from the initial points at time 0 and the trajectories will deterministically follow the same dynamics according to the ODE.

---

> > ### Author Response · Authors · 2022-11-19
> > **Thank you for your review (continued)**
> >
> >
> > * _Runtime of additional baselines and reason they experience exponential growth. Additional memory consumption concerns._
> >
> > The runtime experiment aims to verify that efficient convolution outperforms methods based on recurrent computation (e.g. RNN and ODE) which form the basis for other VAE-based methods. We think that testing ODE-RNN, Latent ODE, ODE2VAE will be representative of this class of models. We also do not claim speed-up against SaShiMi which is also an S4-based model with efficient convolution implemented, although we do claim better modeling flexibility as shown in generation results. For the trend of the plots, please note that on x-axis the sequence length is also scaling exponentially, which corresponds to exponential growth of runtime. Linear relationship in log-log plot is in general polynomial with power equal to the slope. In this case the resulting relationship is roughly linear. For the inference time, to properly test performance we need to fully populate the GPU, which results in using different batch size for different sequence length. The runtime drop likely comes from having lower batch size for longer sequence while ODE integration package is optimized enough to not overcome the drop in runtime due to batch size. Nevertheless, the general increasing pattern holds as a function of sequence length. For memory consumption, we also did additional experiment and found that with the sequence lengths [ 320, 1280, 5120, 20480] and dimension of $h_t$ fixed at 64 with the same batch size, our model achieves GPU memory consumption [1852.625,  1870.625, 1980.625, 2546.625] MB. We also test how memory consumption varies with different dimensions of $h_t$. For dimension of $h_t$ varying in [16,32,64] with sequence length fixed at 320, GPU memory consumption is [1810.62, 1818.62, 1818.52] respectively, roughly constant.
> >
> >
> > * _Why is FID not used?_
> >
> > Computing FID requires a model reliably trained on a wide range of general sequence data. However, to our knowledge, such a model does not exist on general sequence data which can take wildly different form across domains (e.g. finance, climate, etc.). We therefore follow previous works (e.g. SDEGAN) and resort to calculating difference of marginal distributions at different time points and use classifiers as a proxy for generation quality.
> >
> > [1] Gu, Albert, Karan Goel, and Christopher Ré. "Efficiently modeling long sequences with structured state spaces." arXiv preprint arXiv:2111.00396 (2021).

---

### Official Review · Reviewer_5ELF · 2022-10-24

**Confidence:** 3
**Correctness:** 4
**Technical Novelty And Significance:** 3
**Empirical Novelty And Significance:** 2
**Recommendation:** 5

**Clarity, Quality, Novelty And Reproducibility:**

I have no big concerns about the clarity or writing quality of the paper. I do have concerns about the work’s novelty. Please refer to Strength and Weakeness for more details.

**Strength And Weaknesses:**

Strength:

1. The motivation of introducing a latent variable for S4 model is strong. The model leverages the strong representation power of latent variables and efficient convolutional implementation of S4 Model.
2. The work compares against a broad set of baseline models. The experiment results show strong performance and efficiency of the proposed model.


Weakness:
1. Some important baselines are missing from the comparison including VRNN[1] and latent SDE[2].
2. For efficient convolutional implementation, the dynamics of the latent variable are limited to linear dynamics. It’s not clear if such transition dynamics are going to limit the representation power of the latent state. That’s also why I think models with more flexible transition dynamics like VRNN and latent SDE should be compared.
3. The work reports ELBO in experiments but it is not a tight bound of log-likelihood. IWAE as well as its particle filtering version for sequential latent variable model FIVO are tighter than ELBO and can better reflect how well the generative model fits the data.
4. The technical novelty of the proposed model is incremental. The model can be viewed as a special case of the existing sequential latent variable model or latent extension of the S4 model. This is a minor concern as the efficiency improvement based on convolution could compensate for the lack of novelty.


[1] Chung, Junyoung, et al. "A recurrent latent variable model for sequential data." Advances in neural information processing systems 28 (2015).

[2] Li, Xuechen, et al. "Scalable gradients for stochastic differential equations." International Conference on Artificial Intelligence and Statistics. PMLR, 2020.

**Summary Of The Paper:**

The paper proposed LS4 a sequential latent state space model with the latent state evolving according to a discretized approximation of ODE. The structure of the latent state dynamics induces an efficient convolutional implementation. The model is trained in a VAE framework. Experiment results on challenge datasets including data from stiff systems show strong performance of the model

**Summary Of The Review:**

I appreciate the extension of S4 models to its latent variant which leverages both the representation power of sequential latent variable models and the efficiency of S4 models. But I'm especially concerned about the lack of comparison against models with more flexible latent variable models like VRNN and the recently proposed latent SDE.

---

> ### Author Response · Authors · 2022-11-18
> **Thank you for your review and suggestions.**
>
> Thank you for your review and suggestions. Here we address the concerns below:
> 1. _Compare with VRNN and Latent SDE._
>
> We have included additional generation results using VRNN and Latent SDE. Our additional experiments show that LS4 outperforms the two baselines in almost all metrics. For example, on NN5 Daily, VRNN and Latent SDE achieve 0.151 and 0.125 marginal scores (vs. 0.00671 for LS4), 0.00176 and 0.601 classification score (vs. 0.636 for LS4), and 1.22 and 0.957 prediction score (vs. 0.241 for LS4). We show the complete results in the newly added Table 3 in Appendix D.1. Note that although Latent SDE outputforms our model in classification score for Temperature Rain dataset, their output samples (Figure 4 in Appendix D.1) do not resemble ground-truths (Figure 7 in Appendix E). Their high marginal score is indicative of a bad fit to the data. We leave further discussions in Appendix D.1.
>
> 2. _Why do we use linear transition dynamics?_
>
> We stress that the input and output of only a single prior layer follow linear dynamics. We also mention after Equation 11 that we stack multiple such blocks with nonlinearity in between such that the resulting operator is no longer linear, which drastically increases modeling flexibility. More architectural details can be found in Appendix C.
>
> 3. _Does IWAE give tighter bound?_
>
> We additionally implemented IWAE on our model and show that the lower bound indeed becomes much tighter. As shown in the new Table 4 in Appendix D, without further finetuning our model trained with IWAE with 5 particle samples achieves IWAE bound on average 7% better than ELBO.
>
> 4. _Technical Novelty?_
>
> The main technical contribution is two-fold, (1) the introduction of latent variable as a sequence parameterized as a deep stack of SSMs and (2) the efficient parametrization of latent state-space layers as convolutions. These choices lead us to a new design of inference as well as generative paths of our model. For example, we show how to perform fast sampling from the approximate posterior on the latent variables $z_t$ by factorizing it as a product of marginals $p(z_t | x_{<= t})$, which allows us to sample in parallel (across the sequence length) using convolution rather than autoregressively. We also show how introduction of latent variable subsumes vanilla S4 models and experimentally demonstrate its effectiveness in modeling general sequences.

---

### Author Response · Authors · 2022-11-18
**Summary of Responses and Rebuttal Changes**

We thank all the reviewers for their detailed and thoughtful comments. We are grateful for the positive affirmations regarding our model’s motivation and experimental performance. There are, however, several major points made unclear and left to be clarified. We address each reviewer individually and we note some changes to the main text as suggested by the reviewers:

* We additionally compare with VRNN and Latent SDE and add Table 3 and Figure 4, Figure 7 in Appendix D explaining our findings. Our model is still able to outperform the two additional baselines in terms of generation quality.
* A new Table 4 is added to Appendix D comparing models trained with IWAE objective instead of ELBO for tighter fit. On interpolation and extrapolation tasks IWAE’s bounds are consistently lower than ELBO.
* Additional trials results are reported in a separate page in the supplementary zip. Due to time limit we could only test against high efficiency baseline SaShiMi. We show are results outperform the baseline by ~22 standard deviation across datasets for Marginal scores, and is consistently better in other metrics.
* We additionally clarify hyperparameter choices in an additional section in Appendix D.
Figure for the FLAME problem is replotted as suggested.
* We modify the mistakes in Equation 5 and confirm that this is the objective. We connect this objective and further explain at the end of Section 4.3 why we factorize our posterior as a product of marginals.
* We add additional transitioning sentences after Equation 9 for better understanding.
Initialization scheme is clarified in Section 4.4.
* CRPS scores are calculated for probabilistic forecasts as suggested, and we add a new Table 5 in Appendix D and explain our findings. We show that our model is superior than baselines in terms of CRPS scores.

With each reviewer’s suggestions, we have further clarified our model formulation and expanded our experiments to test a wider variety of settings, all of which confirm the expressive modeling power and efficiency of our proposed method. We believe your comments have greatly helped us improve our manuscript and we are happy to answer any additional questions as you see fit.

---

### Decision · Program_Chairs · 2023-01-20

**Decision:**

Reject

**Justification For Why Not Higher Score:**

The reviews for this paper were consistently low and the authors were not able to upgrade their scores sufficiently after rebuttal.

**Justification For Why Not Lower Score:**

N/A

**Metareview: Summary, Strengths And Weaknesses:**

The papers discusses a latent state-space model that evolves according to an approximate discretised ODE. This allows for an efficient implementation using convolutions and training via VAE.  The main challenge the reviewers faced was with the clarity of the paper, with some technical issues in the original submission and concerns that the work appears somewhat incremental. Whilst these may have been addressed to some extent, the overall reviewer sentiment remained that the paper isn't yet strong enough to merit acceptance.